# Development and In Vitro Characterization of [^3^H]GMC-058 as Radioligand for Imaging Parkinsonian-Related Proteinopathies

**DOI:** 10.3390/cells14120869

**Published:** 2025-06-09

**Authors:** Andrea Varrone, Vasco C. Sousa, Manolo Mugnaini, Sandra Biesinger, Gunnar Nordvall, Lee Kingston, Ileana Guzzetti, Charles S. Elmore, Dan Sunnemark, Dinahlee Saturnino Guarino, Sjoerd J. Finnema, Magnus Schou

**Affiliations:** 1Department of Clinical Neuroscience, Centre for Psychiatry Research, Karolinska Institutet and Stockholm Health Services, BioClinicum, J4-14, Akademiska Stråket 1, 17176 Solna, Sweden; 2Department of Clinical Neuroscience, Division of Imaging Core Facilities, Karolinska Institutet, Stockholm, Sweden, BioClinicum, J5, Akademiska Stråket 1, 17176 Solna, Sweden; vasco.sousa@ki.se; 3Translational Sciences, Neuroscience & Ophthalmology Discovery Research, AbbVie Deutschland GmbH & Co. KG, Knollstraße 50, 67061 Ludwigshafen, Germany; manolo.mugnaini@abbvie.com; 4Advanced Cell Technologies & Screening, Neuroscience & Ophthalmology Discovery Research, AbbVie Deutschland GmbH & Co. KG, Knollstraße 50, 67061 Ludwigshafen, Germany; sandra.biesinger@abbvie.com; 5AlzeCure Pharma AB, Hälsovägen 7, 14157 Huddinge, Sweden; gunnar.nordvall@alzecure.org; 6Isotope Chemistry, Early Chemical Development, Pharmaceutical Sciences, R&D, AstraZeneca, Pepparedsleden 1, 43183 Mölndal, Sweden; lee.kingston@astrazeneca.com (L.K.); chad.elmore@astrazeneca.com (C.S.E.); 7Medicinal Chemistry, RIA, Biosciences R&D, AstraZeneca, Pepparedsleden 1, 43183 Mölndal, Sweden; ileana.guzzetti@astrazeneca.com; 8Offspring Biosciences, Forskargatan 20 J, 15136 Södertälje, Sweden; dan.sunnemark@offspringbiosciences.se; 9Department of Clinical Neuroscience, Karolinska Institutet, Center for Molecular Medicine, Karolinska University Hospital, 17176 Stockholm, Sweden; 10Department of Radiology, Perelman School of Medicine, University of Pennsylvania, 1012, 231 S. 34th Street, Philadelphia, PA 19104, USA; dinahlee.saturninoguarino@pennmedicine.upenn.edu; 11Translational Sciences, Neuroscience & Ophthalmology Discovery Research, AbbVie, 1 North Waukegan Road, North Chicago, IL 60064, USA; sjoerd.finnema@abbvie.com; 12Precision Medicine Diagnostic Development and HBS Science, AstraZeneca R&D Oncology, AstraZeneca J4:14, BioClinicum, Akademiska Stråket 1, 17176 Solna, Sweden; magnus.schou@astrazeneca.com

**Keywords:** in vitro assay, neurodegeneration, proteinopathy, Lewy bodies

## Abstract

The molecular imaging of α-synuclein (α-syn) pathology in Parkinson’s disease (PD) and related movement disorders is a clinically unmet need. The aim of this study was to discover and characterize in vitro a radioligand for imaging α-syn pathology. A library of 78 small molecules was developed and screened using recombinant α-syn fibrils and brain homogenates from Alzheimer’s disease (AD) donors. The selection criteria were as follows: *K*i_α-syn_ < 30 nM, *K*i_tau_ and *K*i_A-β_ > 200 nM. Three compounds, GMC-073 (*K*_iα-syn_: 8 nM), GMC-098 (*K*i_α-syn_: 9.7 nM), and GMC-058 (*K*i_α-syn_: 22.5 nM), fulfilled the criteria and were radiolabeled with ^3^H. [^3^H]GMC-058 was the only compound with negligible binding in controls, and was further evaluated using tissue microarrays, autoradiography on fresh-frozen brain slices, and in vitro saturation binding assay on brain homogenates. [^3^H]GMC-058 binding co-localized with α-syn inclusions in Parkinson’s disease (PD) and multiple-system atrophy (MSA), with dense A-β plaques in cerebral amyloid angiopathy and AD and with p-tau inclusions in progressive supranuclear palsy (PSP) and corticobasal degeneration (CBD). Specific binding was highest in PSP and CBD. In vitro *K*_D_ was highest in AD (5.4 nM), followed by PSP (41 nM) and CBD (75 nM). The *K*_D_ in MSA, PD, and controls was >100 nM. [^3^H]GMC-058 is a novel radioligand displaying a low affinity for aggregated α-syn in tissue, with an in vitro profile also suitable for detecting tau pathology in 4R tauopathies.

## 1. Introduction

The accumulation of intracellular aggregates of misfolded α-synuclein (α-syn) is a pathological hallmark of neurodegenerative disorders called synucleinopathies [1]. Such disorders include Parkinson’s disease (PD) and dementia with Lewy bodies (DLB), characterized by the accumulation of pathological α-syn in neurons, in the form of Lewy bodies and Lewy neurites [2], and multiple-system atrophy (MSA), characterized by the accumulation of pathological α-syn in glial cells, in the form of glial cytoplasmic inclusions [3].

In the last decade, extensive research and efforts have been devoted to the search and discovery of imaging agents able to detect in vivo pathological α-syn in patients with synucleinopathies using positron emission tomography (PET). Recent clinical studies with newly developed PET tracers [^18^F]ACI-12589 [4], [^18^F]SPAL-T-06 [5], and [^18^F]C05-05 [6] have shown promising results in patients with MSA ([^18^F]ACI-12589 and [^18^F]SPAL-T-06) and in patients with Lewy-body disease ([^18^F]C05-05). However, these PET radioligands present some shortcomings (Table 1).

[^18^F]ACI-12589 displays in vitro selectivity for α-syn vs. tau and Aβ, and a *K*_D_ in PD and MSA tissue ~30 nM. [^18^F]ACI-12589 is able to successfully image α-syn pathology only in MSA, likely because the *B*_max_/*K*_D_ of the radioligand is not adequate enough to detect a lower level of pathology in PD. In the case of [^18^F]SPAL-T-06 and [^18^F]C05-05, an IC_50_ = 2–3 nM in brain homogenates from DLB or MSA cases has been reported using homologous competition binding studies. However, both radioligands have been developed from the same scaffold as PBB3 and also display an affinity to Aβ and tau. Given that concomitant Aβ and tau pathology is frequent in Lewy-body disease, particularly DLB and PD with dementia, selectivity for α-syn over Aβ and tau is a requirement for a fit-for-purpose α-syn PET radioligand. All clinical PET radioligands developed so far lack the properties of combined high *B*_max_/*K*_D_ and selectivity for α-syn, required to image α-syn pathology in Lewy-body disease with high specificity.

Therefore, a selective PET radioligand able to image α-syn in all synucleinopathies is still a clinically unmet need. The aim of this study was to discover small molecules displaying suitable in vitro profiles (affinity in nM range and >10-fold selectivity for α-syn vs. Aβ and tau) for development as α-syn PET radioligands. The work included in vitro screening using recombinant α-syn fibrils and brain homogenates from Alzheimer’s disease (AD) donors. Compounds with suitable in vitro affinity profiles were radiolabeled with ^3^H. Potential candidates were evaluated with in vitro autoradiography (ARG). Brain tissue from control donors and from patients with different proteinopathies was used for ARG experiments using fresh-frozen tissue sections and paraffine-embedded tissue microarrays (TMAs). High-resolution ARG and immunohistochemistry on consecutive TMA sections were performed to study the co-localization of an ARG signal with α-syn, p-tau and Aβ pathology.

## 2. Materials and Methods

### 2.1. In Vitro Binding Assays

#### 2.1.1. Radioligand Binding to α-Syn Fibrils

Compound affinity for α-syn fibrils was measured by means of radioligand binding displacement experiments of an AbbVie tool radioligand with high affinity for α-syn fibrils ([^3^H]-α-syn-tool). The aggregation of α-syn monomers into fibrils was obtained as described by Bousset et al. [7]. In brief, monomers (5 mg/mL, prepared in AbbVie (Worcester, MA, USA) were incubated in 150 mM of KCl and 50 mM of Tris-HCl (pH 7.5) for 7 days at 37 °C, under constant shaking conditions (650 rpm, Eppendorf Thermomixer C, Eppendorf, Hamburg, Germany). After incubation, the fibrils were harvested via ultra-centrifugation (40,100× *g*, 30 min, 20 °C), resuspended in buffer, sonicated (40% amplitude, 10 s pulse on, 5 s pulse off) for 4.5 min at 20 °C, divided into aliquots (5.5 mg/mL), and stored at −80 °C until the day of the experiment. The characterization of the fibrils included protein content determination (BCA method) and an analysis of Proteinase K degradation pattern [7].

On the day of the experiment, α-syn fibrils were thawed for a few minutes at 37 °C, diluted in Dulbecco’s Phosphate Buffered Saline (PBS) without calcium and magnesium, pH = 7.4 (DPBS) plus 0.1% bovine serum albumin (BSA), and incubated (at the final fibrils concentration of 2.5 µg/mL) for 60 min, at room temperature (RT), with 5 nM [^3^H]-α-syn-tool, in the absence or presence of different concentrations of non-radiolabeled compounds. Bound radioligand was separated from free radioligand via rapid filtration and washings with cold DPBS, using a Unifilter-96 GF/B filter plate (PerkinElmer, Rodgau, Germany) presoaked in 0.3% polyethylenimmine (PEI) and a Unifilter Cell Harvester (PerkinElmer, Rodgau, Germany). Plates were air-dried, and the scintillation liquid (Betaplate Scint, PerkinElmer) was added and counted using a MicroBeta2 2450 (Perkinelmer, Rodgau, Germany). Non-specific binding (NSB) was defined as the binding in the presence of 10 µM unlabeled α-syn tool. Data fitting and the constant of inhibition (*K*i) calculation was performed using the software platform from Dotmatics (Boston, MA, USA).

#### 2.1.2. Radioligand Binding to Native Tau and Aβ Fibrils

Compounds’ affinity for tau and Aβ fibrils was measured by means of radioligand binding displacement experiments on AD brain homogenates, using [^3^H]NFT-355 [8] (AbbVie, North Chicago, IL, USA) and [^3^H]Pittsburg Compound-B (PiB; Novandi Chemistry AB, Södertälje, Sweden [9]) to label tau and Aβ aggregates, respectively. Flash-frozen tissue from the frontal cerebral cortex of an AD (Braak stage V) brain was purchased from Analytical Biological Services Inc. (ABS; Wilmington, DE, USA), homogenized in DPBS, divided into aliquots (10 mg/mL), and stored at −80 °C until the day of the experiment.

On the day of the experiment, the brain homogenate was brought to RT, diluted in DPBS plus 0.1% BSA and incubated for 90 min, at RT, with 0.5 nM [^3^H]NFT-355 or 1 nM [^3^H]PiB, in the absence or presence of different concentrations of non-radiolabeled compounds. Bound radioligand was separated from free radioligand via rapid filtration and washings with cold DPBS, using a Unifilter-96 GF/B filter plate (presoaked in 0.3% PEI) and a Unifilter Cell Harvester (PerkinElmer). Plates were air-dried, and the scintillation liquid (Betaplate Scint, PerkinElmer) was added and counted using a MicroBeta2 2450. NSB was defined as the binding in the presence of 10 µM T808 [10] (AbbVie Germany) and 10 µM PiB for [^3^H]NFT-355 and [^3^H]PiB, respectively. Data fitting was performed using the software platform from Dotmatics (Boston, MA, USA).

### 2.2. Radiosynthesis of [^3^H]GMC-058, [^3^H]GMC-073, and [^3^H]GMC-098

Chemicals and solvents were obtained from commercial sources and were used without further purification. NMR spectra were recorded on a Bruker 500 MHz AVANCE III system using standard Bruker pulse sequences. Experiments were run in D6-DMSO at 25 °C. ^1^H NMR chemical shifts were referenced relative to the residual solvent peak at 2.50 ppm, and ^13^C NMR chemical shifts were referenced to 39.5 for D6-DMSO. Flash column chromatography was carried out using prepacked silica gel columns supplied by Biotage and using a Biotage automated flash system with UV detection. Preparative HPLC was performed using a Waters Xbridge C18 5 µ OBD 19X150 mm, 25 mL/min, 30% for 2 min and then was ramped to 95% over 18 min and held for 3 min MeCN-0.2% aq. NH4OH was used unless otherwise indicated. Reactions with tritium gas were performed on an RC Tritec tritium manifold. Analytical HPLC was carried out using an Agilent 1100 series HPLC using a Waters 4.6 × 100 mm Xbridge C18 3.5 μ, with the following elution profile: 5% for 3 min and then ramped to 95% over 22 min and hold 95% for 5 min MeCN/water 10 mM NH_4_HCO_3_ pH 10 with radioactive detection using a LabLogic Beta-Ram 4 detector. LCMS analysis was carried out using a Waters 1100 HPLC system with a Waters 3100 mass detector on an XSelect CSH C-18 4.6/150 mm, 3.5 μm with a gradient of 5–100% MeCN–0.1% formic acid (adjusted to pH 3) over 10 min followed by a 2 min wash with 100% MeCN or on a Waters1200 series UPLC with a Waters 4.6 × 100 mm Xselect CSH C18 3.5 μ, 5 to 95% MeCN/water 0.2% formic acid pH 3 for 1.85 min then isocratic elution for 0.1 min with mass detection using a QDA mass detector. The molar activities of the products were determined via LC/MS using Isopat2 to deconvolute the MS signals [11]. Liquid scintillation counting was performed with a Beckman LS 6500 scintillation counter.

### 2.3. Autopsy Material

Frozen human brain tissues from patients with Parkinson’s disease (PD), Lewy-body disease (LBD), multiple-system atrophy (MSA), corticobasal degeneration (CBD), and non-dementia controls were obtained from the Netherlands brain bank (Table 2) and frozen human tissues from patients with MSA, PSP, and one control were obtained from the UK brain bank (Table 3). Formalin-fixed paraffine-embedded tissue blocks from patients with different synucleinopathies were obtained from the Netherland Brain Bank (Table A1). Approval for the use of the autopsy material for the project was obtained from the Swedish Ethical Research Authority. Brain homogenates were obtained from tissue of AD, CBD, PSP, MSA-P patients, and healthy controls, obtained from the Alzheimer’s Disease Research Center of the University of Pittsburgh (Table A2), from the University of California San Francisco (Table A3), and from the Banner Sun Health Research Institute (Table A4).

### 2.4. Preparation of Human Brain Tissue for In Vitro Binding Studies

As described in Bagchi et al. [12], to prepare insoluble fractions from PD and MSA-P patients, fresh-frozen tissue blocks were sequentially homogenized in four buffers (3 mL/g wet weight of tissue) with glass Dounce tissue grinders (Kimble, Vineland, NJ, USA): (1) high salt (HS) buffer: 50 mM Tris-HCl pH 7.5, 750 mM NaCl, 5mM EDTA; (2) HS buffer with 1%Triton X-100; (3) HS buffer with 1%Triton X-100 and 1M sucrose; and (4) phosphate-buffered saline (PBS). Homogenates were centrifuged at 100,000× *g* after each homogenization step, and the pellet was resuspended and homogenized in the next buffer in the sequence. For AD, CBD, PSP, and CT cases, the frozen tissue blocks were prepared as described by Stehouwer et al. [13]. In brief, tissue blocks were thawed and homogenized in ice-cold pH 7.0 phosphate-buffered saline (PBS) at 300 mg/mL on ice using a glass homogenizer, diluted 30-fold with PBS to 10 mg/mL, and homogenized a second time with a Brinkmann Polytron homogenizer before storage at −80 °C.

### 2.5. Saturation Binding Assays

Saturation binding assays were performed in AD, CBD, PSP, MSA-P, PD, and healthy patients brain homogenates (0.1 mg/mL tissue) using a [^3^H]GMC-058 concentration range of 0.9 nM to 80 nM and incubation for 90 min at room temperature (RT) with PBS + 20% EtOH. Non-specific binding was determined using 10 μM of unlabeled GMC-044. After incubation, samples were filtered under vacuum on equilibrated GF/B UniFilter plates (PerkinElmer) using the FilterMate 196 (PerkinElmer). Afterwards, filters were washed three times with 250 μL chilled buffer (PBS + 20% EtOH). [^3^H]GMC-058 binding was measured with a beta scintillation counter (PerkinElmer). The saturation data was fitted and analyzed using the non-linear regression function of GraphPad Prism 10 software to calculate the dissociation constant (*K*_D_) and maximum number of binding sites (*B*_max_). Scatchard plots were prepared with GraphPad Prism 10 software to display the saturation binding data.

### 2.6. Autoradiography Experiments on Fresh-Frozen Tissue Sections

Selections of frozen human sections (described in Table 2 and Table 3) were used for binding autoradiography. Sections were first pre-incubated for 20 min in binding buffer [50 mM Tris-HCl (pH 7.4), 120 mM NaCl, 5 mM KCl, 2 mM CaCl_2_, 1 mM MgCl_2_, and 0.1% BSA] and then incubated with [^3^H]GMC-058 (specific activity 22 Ci/mmol); [^3^H]GMC-073 (specific activity 23.4 Ci/mmol); or [^3^H]GMC-073 (specific activity 26.4 Ci/mmol) in binding buffer for 180 min at RT. In the pilot autoradiography experiment (Figure A1). [^3^H]GMC-058 was tested at 25 and 50 nM, while [^3^H]GMC-073 and [^3^H]GMC-098 were each tested at 10 and 20 nM. For all other autoradiography in fresh-frozen tissue sections, 25 nM [^3^H]GMC-058 was used. To determine NSB, adjacent brain sections were incubated with [^3^H]-tracer mixed with 5 μM of unlabeled GMC-044. The binding reaction was stopped with two 10 min washes in washing buffer [50 mM Tris-HCl (pH 7.4) at 4 °C] and a brief dip in cold distilled water. Slides were allowed to air-dry before being placed under storage phosphor screens (Fujifilm Plate BAS-TR2025, Fujifilm, Tokyo, Japan) in imaging cassettes for 90 h together with ART0123C and ART0123B tritium standards on glass slides (American Radiolabeled Chemicals Inc., St. Louis, MO, USA). Storage phosphor screens were scanned using an Amersham Typhoon FLA-9500 phosphor imaging scanner (Cytiva, Marlborough, MA, USA), and the resulting images were analyzed using Multi Gauge 3.2 phosphor imager software (Fujifilm, Tokyo, Japan) for ROI delimitation, density calibration, and quantification. Quantitative binding data from duplicates of the total, and NSB were plotted and analyzed using graphPad Prism v10 (GraphPad Software, Boston, MA, USA). Specific binding was determined by subtracting the non-specific signal from the total signal. The toluidine blue staining of adjacent sections was used to obtain anatomical references for white- and gray-matter ROI selection. Care was taken not to include artifact signals in the ROIs analyzed. These artifacts were identified as randomly placed punctate signals appearing in both total and NSB images, and they often coincided with small folds or tears in the tissue sections.

### 2.7. Autoradiography and Emulsion Autoradiography on Tissue Microarrays

Tissue micro-arrays (TMAs), prepared from paraffin-embedded tissue from human post-mortem tissue as described in [14], were first de-paraffinized with two 1 h and one overnight incubation in Neo-Clear Xylene Substitute (Sigma-Aldrich, St. Louis, MO, USA) at 37 °C and then dehydrated in three short incubations at RT with decreasing concentrations of ethanol before autoradiography with [^3^H]GMC-058 was performed as described above. For the saturation binding autoradiography, duplicate slides of each TMA set were incubated with 0.3 nM to 200 nM [^3^H]GMC-058. Adjacent sections were co-incubated with 5 μM of unlabeled GMC-044 to calculate NSB. The quantification of total and non-specific binding was performed via delineation of the ROIs on the whole core, except for MSA, where the ROI followed the areas in which the binding of [^3^H]GMC-058 corresponded to areas with α-syn inclusions. Those cores that had imperfections or were incomplete were not included in the analysis, or only the area of remaining tissue that adhered to the glass slide was delineated. For the cores from the substantia nigra of PD patient cases, the cores selected for analysis were those containing α-syn inclusion, but no A-β deposits of pTau tangles. The information was collected from IHC analysis of adjacent TMA slides.

For high-resolution emulsion autoradiography, an autoradiography experiment with TMAs using 25 nM [^3^H]GMC-058 was performed as described above. At the end of the 90 h exposure to the storage phosphor screen, slides were stored in a vacuum until dipping in a photoemulsion. In the dark, using a sodium lamp, Agar Scientific, AGP9284 type NTB emulsion was melted in a heated water bath and diluted (1:1) with ddH2O. The emulsion was poured into dipping chambers, and the slides were dipped and placed vertically to air-dry. When completely dry, the slides were placed in a light-tight slide box with desiccant and exposed at 4 C for 1–12 weeks. After exposure, the slides were developed in diluted Phenisol Developer for 2 min, rinsed in water, and fixed in hypam Fixer at 17 °C in a water bath. After rinsing in water, the slides were counterstained with Harris HTX, dehydrated in graded ethanol, cleared in Xylene, and mounted in Pertex. Images were taken using a 3D Histech P250 III scanner (3DHISTECH Kft. Hungary) at up to 15 focal layers with a distance of 0.2 μm.

### 2.8. Immunohistochemistry

Immunohistochemical chromogenic (IHC) staining was made essentially as in [15], and the reference antibodies for the respective proteinopathies used were Signet laboratories (Abeta 6E10, Abeta 4G8), Abcam (Anti-Alpha-synuclein antibody [LB 509]), and Thermo Fisher Scientific (Waltham, MA, USA) (AT-8 Phospho-PHF-Tau). Briefly, 5 μm slide-mounted tissue sections were performed according to a standardized protocol with modifications to optimize the specificity of the observed staining patterns. IHC staining was performed using the Discovery Ultra platform (Ventana, Roche Diagnostics International AG, Rotkreuz, Switzerland) automated immunostaining robot, using the OmniMap DAB chromogenic staining kit (Ventana Medical Systems) according to the manufacturer’s instructions. In brief, initial deparaffinization, followed by heat-activated antigen retrieval in a pH 8.0 buffer for 92 min (Ventana Medical Systems’ (Ventana) DISCOVERY CC1 (DISCOVERY Cell Conditioning Solution 1, 06414575001, 950–500), was performed to improve the detection of antigens in the FFPE tissue. Endogenous tissue peroxidases (Inhibitor CM, Ventana), which may interfere with the assays, were blocked with 0.3% hydrogen peroxide. The primary antibod forphosphorylated Tau detection (AT-8 pTau [In vitrogen, MN1020] at 2 µg/mL), and re for total Abeta (Mouse anti-Abeta 4G8 [Signet, Covance, 800709 (SIG-39200)], were mixed and co-incubated at a final concentration of 1 µg/mL) and applied on the tissue sections. For the IHC staining of α-synuclein inclusions, the anti-human α-synuclein antibody LB509 (Abcam, GR300971) 0,5 µg/mL was used. Prior to staining, sections were treated for 24 min with protease (Ventanas Protease 1, [760–2018]) to degrade soluble α-synuclein species. Antibodies were followed by incubation with the HRP-conjugated secondary antibody (the HPR labeled OmniMap goat anti-Mouse Ab, 05269652001 760–4310), and the visualization of the positively stained cells was performed through the addition of hydrogen peroxide and DAB (DISCOVERY ChromoMap DAB Kit [Ventana, 760–159) (single IHC)], resulting in an insoluble brown (DAB) staining precipitate at the site of antibody binding. Counterstaining for IHC was performed with hematoxylin [Hematoxylin II, Ventana, 760–2208 and Bluing Reagent, Ventana, 760–2037]. The stained slides were subsequently scanned at up to 10 focal layers in brightfield (20× -40 objective) using a digital whole slide scanner (Pannoramic 250 III Scanner, 3DHistech, Budapest, Hungary). Analysis was performed of the IHC stained tissue sections via manual evaluation using a digital image viewer software (CaseViewer_2.0_RTM_v2.0.2.61392). All analyses of the stained and labeled tissue sections with reference antibodies for Aβ, p-tau, and α-syn and/or ligands were compared, side by side, to each other. Adjacent fresh-frozen tissue sections or TMAs were used when comparing IHC immunoreactivity with an autoradiography binding signal.

## 3. Results

The work plan for this study included a first part that was devoted to the discovery of small molecules with an in vitro profile suitable for the development of a selective α-syn ligand and a second part that included the in vitro characterization of lead candidates.

The discovery part included compound design and in vitro competition experiments for the identification of lead candidates. The in vitro characterization included ^3^H-radiolabeling, in vitro saturation binding, and autoradiography experiments in tissues from donors with different proteinopathies.

### 3.1. Discovery and In Vitro Characterization of GMC-058

#### 3.1.1. Discovery of GMC-058, GMC-073 and GMC-098

A library of 78 small molecules was generated based on available information from the public domain. Four optimization cycles were performed with radioligand binding assays to measure the affinity of compounds for in vitro-assembled α-syn fibrils and native tau and Aβ fibrils in AD homogenates.

Sixteen of these compounds showed affinity for α-syn fibrils < 30 nM and for Aβ and tau > 50 nM (Table A5). The criteria for selection were *K*i for α-syn < 30 nM and *K*i for tau and Aβ > 200 nM.

Three compounds, GMC-058, GMC-073, and GMC-098, were selected based on those criteria and on the feasibility of radiolabeling. The *K*i of GMC-058, GMC-073, and GMC-098 using recombinant α-synuclein fibrils was 22.5 nM, 8 nM, and 9.7 nM, respectively.

All three compounds showed a lower affinity for Aβ (*K*i = 1490 nM for GMC-058, 2630 nM for GMC-073, and 226 nM for GMC-098) and tau (*K*i= 1320 nM for GMC-058, 248 nM for GMC-073, and 805 nM for GMC-098) in AD brain homogenates. GMC-058 (Figure 1), GMC-073, and GMC-098 were, therefore, selected for radiolabeling with ^3^H.

#### 3.1.2. Autoradiography Experiments with [^3^H]GMC-058, [^3^H]GMC-073 and [^3^H]GMC-098

The three [^3^H]-labeled compounds were initially evaluated through autoradiography in tissue sections from the cingulate cortex of healthy controls.

[^3^H]GMC-073 and [^3^H]GMC-098 displayed clear binding in control cingulate cortex tissue, approximately 50% of which was displaced via 5 µM GMC-044, whereas [^3^H]GMC-058 displayed negligible binding (Figure A1 and Table A6). The nature of this off-target displaceable binding is not known. [^3^H]GMC-058 was selected for further evaluation in pathological tissue sections.

#### 3.1.3. In Vitro Saturation Binding Assays with [^3^H]GMC-058 in Brain Homogenates

Saturation binding assays with [^3^H]GMC-058 were conducted in brain homogenates from cases with AD, PSP, CBD, MSA-P, PD, and Controls (Figure 2 and Figure A2).

The *K*_D_ estimated in AD tissue was 5.4 nM, and in PSP tissue, it was 46 nM, while in CBD tissue, it was 71 nM. In brain homogenates from MSA-P and control cases, the *K*_D_ was >100 nM. No evidence of saturation binding was observed in PD tissue.

#### 3.1.4. Autoradiography Experiments with [^3^H]GMC-058

Autoradiography experiments using 25 nM [^3^H]GMC-058 in tissue sections from the cingulate cortex of controls and PD patients and from the cerebellum of MSA patients did not show clear evidence of specific binding higher than the displaceable (off-target) binding in controls (Figure 3 and Figure A3 and Table A7). Saturation [^3^H]GMC-058 binding autoradiography experiments in TMAs showed that specific binding did not reach saturation in any of the tissues (Figure A4 and Figure A5) and did not permit us to obtain reliable *K*_D_ estimates.

However, the specific binding data obtained at the concentration of 25 nM [^3^H]GMC-058 showed clear differences in specific binding and were thus selected for group comparison.

In synucleinopathies (PD and MSA), cerebral amyloid angiopathy (CAA), AD, and 4R tauopathies (PSP and CBD), the specific binding of 25 nM [^3^H]GMC-058 was significantly higher than the displaceable binding in controls (*p* < 0.01, ANOVA with Dunnett’s multiple comparisons test; see Table 4 and Figure A6). High-resolution autoradiography showed that the binding of 25 nM [^3^H]GMC-058 co-localized with α-syn inclusions in PD and MSA, with dense Aβ plaques in CAA and AD, and with p-tau inclusions in PSP and CBD (Figure 4, Figure A7, Figure A8 and Figure A9).

We then conducted autoradiography experiments in fresh-frozen tissue for which α-syn, Aβ, and p-tau expression was measured via IHC. We selected tissue sections of cerebellum from two donors with MSA, of superior frontal gyrus from one donor with CBD, of globus pallidus (GP) from two donors with PSP, and of the cingulate cortex, caudate, GP, and putamen from two controls (Figure 5 and Figure A10 and Table A8).

In the two MSA cases, specific binding measured in cerebellar gray matter was two-fold higher than the average displaceable binding measured in the gray matter of the cingulate cortex, caudate, the globus pallidus, and the putamen of the controls (Figure 5A,D,E). In CBD and PSP cases, the specific binding in the gray matter of the superior frontal gyrus and the globus pallidus was, on average, three-fold and ten-fold higher, respectively, than the displaceable binding measured in controls (Figure 5B,C,E). No specific binding was observed in the white matter of control cases, whereas in MSA and CBD cases, the average specific binding was 35 to 80 fmol/mg (Figure 5A,B,E and Table A8).

## 4. Discussion

This study was designed to identify a small molecule with in vitro properties suitable for development as ligand binding to pathological α-syn. We identified three compounds, GMC-058, GMC-073, and GMC-098, that displayed selective binding to recombinant α-syn fibrils. The three compounds were radiolabeled with ^3^H, and among the three compounds, [^3^H]GMC-058 was selected since it showed the lowest non-displaceable binding in control tissue. However, when [^3^H]GMC-058 was further evaluated in human brain tissue homogenates and sections, it displayed low specific binding in synucleinopathy cases but clear specific binding in PSP and CBD cases. This observation was unexpected, considering that the *K*i of GMC-058 measured in recombinant α-syn fibrils was 22.5 nM, and in tau-enriched AD tissue was >200 nM. It is possible that the affinity of GMC-058 measured using recombinant α-syn fibrils was different from the affinity of [^3^H]GMC-058 to native α-syn fibrils in tissue slices or brain homogenates, resulting in a discrepancy between the two different assay results. With regards to tau, since the competition studies were performed using the reference tau tracer [^3^H]NF-355, it is possible that GMC-058 recognizes a different binding site on tau fibrils. Several radioligands have been developed for imaging tau pathology [16]. Second-generation tau PET radioligands [^18^F]PI-2620 and [^18^F]PM-PPB3 (also known as [^18^F]APN-1607, or florzolotau) have been studied in patients with AD [17,18] and PSP [19,20]. Those studies showed that both radioligands can image 3R and 4R tau pathology in vivo. More recently, the discovery, in vitro characterization, and in vivo pre-clinical assessment of [^18^F]OXD-2314 has been reported [21], indicating that this radioligand also has the potential to image 3R and 4R tauopathies. Although [^3^H]GMC-058 has been designed and developed as a potential α-syn PET radioligand, the in vitro properties suggest that the *B*_max_/*K*_D_ is not high enough to enable imaging of α-syn pathology in vivo. On the other hand, the evidence of higher specific binding of [^3^H]GMC-058 in PSP and CBD than in MSA, suggests that GMC-058 might have potential as a PET radioligand for imaging 4R tauopathies. A discrepancy was observed between the high *K*_D_ (>50 nM) of [^3^H]GMC-058 in brain homogenates from PSP and CBD cases and the clear evidence of specific binding in fresh-frozen brain sections from CBD and PSP cases. A similar discrepancy between the results of in vitro saturation studies in brain homogenates and autoradiography studies on brain sections has been observed for another potential α-syn tracer, [^3^H] -4i, and is difficult to explain [22].

## 5. Conclusions

[^3^H]GMC-058 is a novel radioligand displaying a low in vitro affinity for aggregated α-syn, with an in vitro profile also suitable for imaging tau pathology in 4R tauopathies. Further evaluations, including radiolabeling with ^11^C and in vivo imaging in non-human primates, are warranted to further assess the potential of GMC-058 as a potential PET radioligand for 4R tau.

## Figures and Tables

**Figure 1 cells-14-00869-f001:**
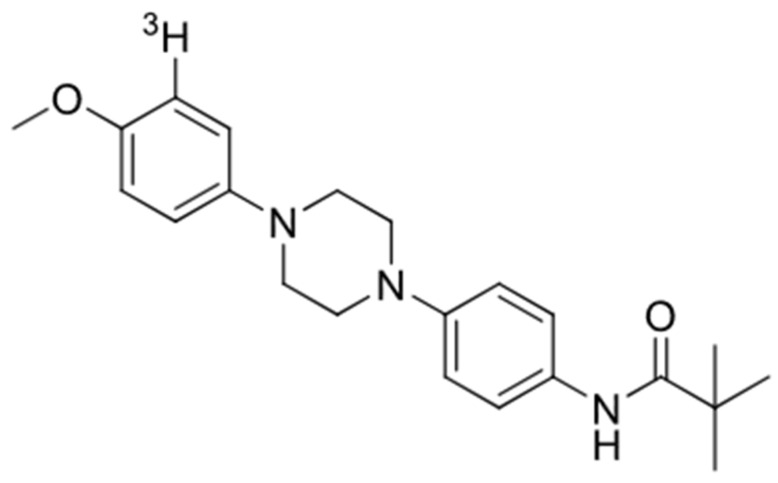
Chemical structure of [^3^H]GMC-058.

**Figure 2 cells-14-00869-f002:**
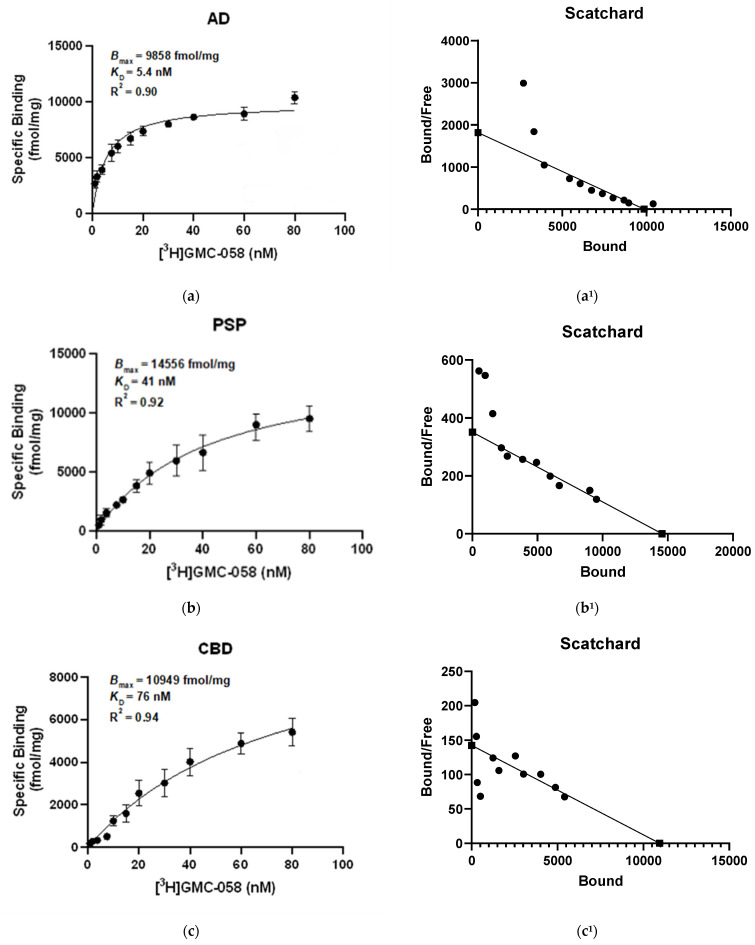
[^3^H]GMC-058 saturation binding assays in (**a**,**a^1^**) AD, (**b**,**b^1^**) PSP, (**c**,**c^1^**) CBD, (**d**,**d^1^**) MSA-P, (**e**) PD and (**f**,**f^1^**) control brain homogenates. [^3^H]GMC-058 saturation binding assays were carried out in brain tissue homogenates from AD, PSP, CBD, MSA-P, PD, and control cases using concentrations of 0.9 to 80 nM. Non-specific binding was determined using 10 μM of unlabeled GMC-058. Scatchard plots indicating *B*_max_ and *K*_D_ values. Error bars represent the mean ± SD for two experiments in triplicates. *B*_max_ = maximum number of binding sites; *K*_D_ = dissociation constant.

**Figure 3 cells-14-00869-f003:**
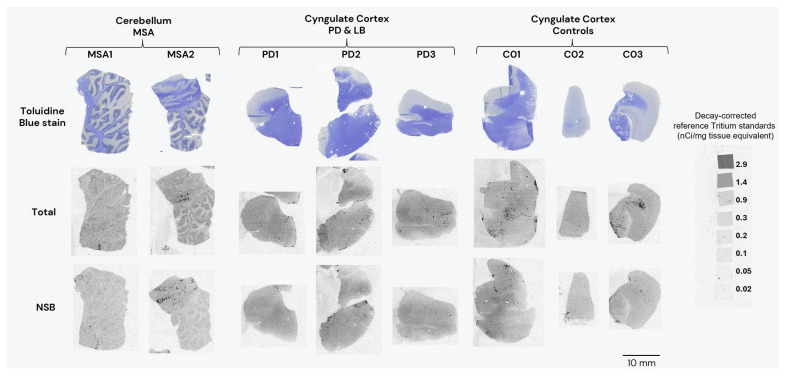
Representative images from autoradiography with [^3^H]GMC-058 (25 nM) on fresh-frozen tissue sections from patients with multiple-system atrophy (MSA), Parkinson’s disease (PD), Lewy-body disease (LB), and controls. Non-specific binding was measured with excess (5 µM) of blocker (GMC-044). Line scale: 10 mm.

**Figure 4 cells-14-00869-f004:**
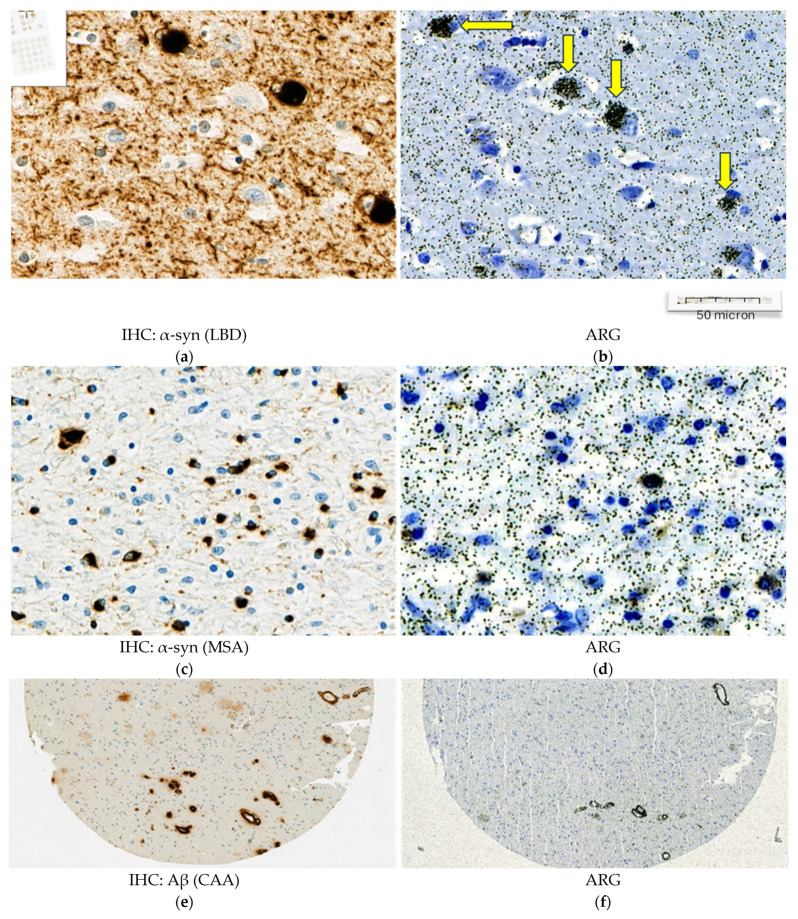
Representative high magnification images from α-synuclein (α-syn), amyloid beta (Aβ) and, phospho-tau (p-tau) immunohistochemistry (IHC) and emulsion autoradiography (ARG) with 25 nM [^3^H]GMC-058. (**a**–**d**) In two representative LBD (**a**,**b**) and MSA cases (**c**,**d**), the ARG signal (yellow arrows) corresponds to Lewy bodies (**a**) and oligodendrocytic inclusions (**c**). (**e**,**f**) In CAA cases, the ARG signal corresponds to dense perivascular Aβ deposits (**e**). (**g**–**j**) In PSP (**g**,**h**) and CBD (**j**,**k**), the ARG signal seems to localize in areas of p-tau positive inclusions (**g**,**i**). (**k**,**l**) In AD the case with abundant Aβ plaques and p-tau inclusions (**k**), a weak ARG signal was observed (**l**).

**Figure 5 cells-14-00869-f005:**
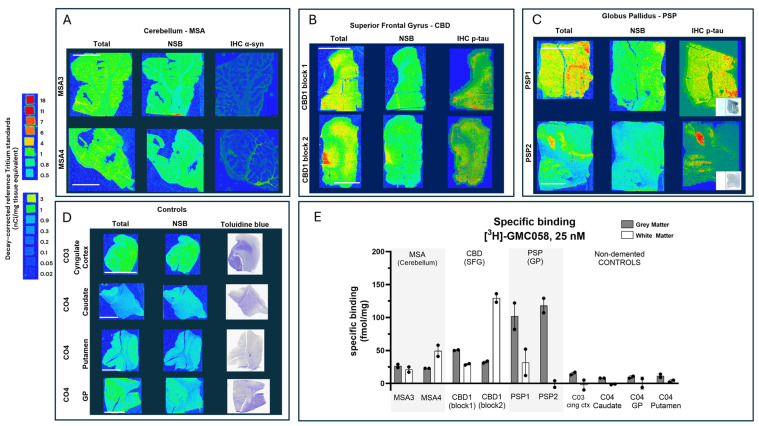
Representative images from autoradiography with [^3^H]GMC-058 (25 nM) and corresponding immunohistochemistry (IHC) heatmap showing the density of α-synuclein (α-syn) or phospho-tau (p-tau) immunoreactivity on fresh-frozen tissue sections from patients with (**A**) multiple-system atrophy (MSA); (**B**) cortical-basal degeneration (CBD); (**C**) progressive supranuclear palsy (PSP); and (**D**) non-dementia controls. Non-specific binding was measured with excess (5 µM) of cold blocker (GMC-044). (**E**) Specific binding (fmol/mg) quantified from gray and white matter regions of interest in all cases. Bars represent the average ± s.e.m. of two technical replicates quantified from each case. Line scale: 10 mm. C04 was negative for pTau, α-Syn, and Aβ; therefore a Toluidine blue stain is shown in (**D**).

**Table 1 cells-14-00869-t001:** Main in vitro and in vivo characteristics of α-syn PET radioligands evaluated in patients with synucleinopathies.

Radioligand	IC_50_ or *K*_D_			Synucleinopathies Evaluated **
	α-syn	Aβ	pTau	
[^18^F]ACI-12589	33.5 ± 17.4 nM (sporadic PD); 28 nM (MSA)	317 nM (AD)		PD ***, MSA (+), DLB
[^18^F]SPAL-T-06	2.5 nM (MSA)	Not reported		MSA (+)
[^18^F]C05-05	IC50 *: 1.5 nM (DLB);	IC50 * in AD tissue: 12.9 nM		PD/DLB (+) and MSA (+)

* = only 50% of the total binding was displaced in homologous competition experiments. ** = (+) indicates the cases in which specific binding higher than controls was measured. *** = two participants with duplications of the SNCA gene were included in the study.

**Table 2 cells-14-00869-t002:** Demographic data of cases from the Netherlands Brain Bank selected for autoradiography studies using fresh-frozen tissue.

Case n.	Diagnosis	Age	Sex	PMI (h)	Brain Region
PD1	Parkinson’s disease	69	M	7	Cingulate gyrus
PD2	Parkinson’s disease	77	M	5	Cingulate gyrus
PD3	Parkinson’s disease	75	M	6	Cingulate gyrus
LBV1	Lewy-bodies variant	83	M	5	Cingulate gyrus
LBV2	Lewy-bodies variant	54	M	5	Cingulate gyrus
MSA1	Multi-system atrophy	66	M	5	Cerebellum
MSA2	Multi-system atrophy	69	F	5	Cerebellum
CBD1	Corticobasal degeneration	58	F	7	Superior frontal gyrus
C01	Non-dementia control	81	F	4	Cingulate gyrus
C02	Non-dementia control	51	M	8	Cingulate gyrus
C03	Non-dementia control	79	F	11	Cingulate gyrus

**Table 3 cells-14-00869-t003:** Demographic data of cases from the UK Brain Bank selected for autoradiography studies using fresh-frozen tissue.

Case n.	Diagnosis	Age	Sex	PMI (h)	Brain Region
MSA3	Multiple-system atrophy	58	F	10	Cerebellum
MSA4	Multiple-system atrophy—cerebellar form	52	M	33	Cerebellum
PSP1	Progressive supranuclear palsy	66	M	8	Globus pallidus
PSP2	Progressive supranuclear palsy	77	F	11	Globus pallidus
C04	Control	68	M	7.5	Caudate, putamen and globus pallidus

**Table 4 cells-14-00869-t004:** Specific binding (fmol/mg tissue equivalent) of 25 nM [^3^H]GMC-058, measured in TMAs from cases with different proteinopathies. In non-dementia controls (NDE) and liver tissue, the values refer to displaceable (off-target) binding.

TMAs	NDE	Liver Control	MSA	CAA	AD	CBD	PSP	PD
*N*=	8	4	1	3	4	2	2	6
Mean (±SEM)	70	57	166 **	131 **	127 **	204 ***	173 ***	129 ***
	(4)	(7)	(19) †	(12)	(15)	(38)	(2)	(7)

NDE = non-dementia elderly controls; MSA = multiple-system atrophy; CAA = cerebral amyloid angiopathy; AD = Alzheimer’s disease; CBD = corticobasal degeneration; PSP = progressive supranuclear palsy; PD = Parkinson’s disease. *** *p* < 0.0001; ** *p* < 0.01, compared to NDE control, obtained via one-way ANOVA with Dunnett’s multiple comparisons test. † SEM from 10 technical replicates of n = 1 MSA case.

## Data Availability

The raw data supporting the conclusions of this article will be made available by the authors upon request.

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
