# Peer review of "Development and In Vitro Characterization of [3H]GMC-058 as Radioligand for Imaging Parkinsonian-Related Proteinopathies"

_cells, 2025, doi:10.3390/cells14120869_

Round 1
Reviewer 1 Report
Comments and Suggestions for Authors
In this study, the authors present the identification and characterization of [3H]GMC-058 for imaging pathological alpha-synuclein aggregates. They use saturation binding and in-vitro autoradiography assays to characterize the binding of this radiotracer. The authors have attempted characterization of the radiotracer in a variety of disease and disease-control tissues, an important experimental paradigm in the pre-clinical validation of novel radiotracers. This tracer showed negligible binding in Parkinson’s Disease and Multiple System Atrophy tissues. Interestingly, the authors found that the tracer shows binding in Progressive supranuclear palsy (PSP) and Corticobasal degeneration (CBD) tissues, which are both characterized by the accumulation of 4R tau. The authors also observed discrepancies in binding between tissue homogenate binding assays vs. autoradiography in tissue sections, indicating complexity and variability in the methods of radiotracer assessment. Due to binding seen in PSP and CBD, the tracer may have potential to visualize 4R taupathies in vivo.
Most data is presented by quantification of autoradiography data, which in itself can be challenging. In addition, there is a real potential of operator bias in deciding where to put the ROIs. Therefore, more clarity is needed to improve the manuscript. At this stage, the manuscript needs major revision before publication:
Figure 5 is missing from the reviewer version, so that data has not been reviewed in this report. Please ensure that it is present in revised version.
Another major issue for review are the low resolution figures in the reviewer version. For example, the Scatchard plots in Figure 2 are blurred and un-intelligible. Is there a second binding site in the Scatchard?
It would be helpful to use the global fit analysis to analyze total and non-specific binding at the same time. This helps to visualize the issues with non-specific binding at the same time as total binding. This may be important as a very high concentration of 10uM is used for competition.
As mentioned before, the authors present substantial quantitative autoradiography analysis. For readers, it would be helpful to detail how the quantification of autoradiography was performed? Please add in supplementary data figures showing where the ROI boxes are placed on representative tissue sections used for quantification. (Line 222).
Similarly, it was unclear how quantification of autoradiography was performed for Appendix B (Line 630).
Similarly, in Table 3, specific binding is quantified from several cases and mean±SEM is presented. Please elaborate on how ROI delineation was done, how total and non-specific ROIs were generated in the TMA experiments? Was the entire core a single ROI? In supplementary figures, show representative sections with ROIs to show how measurements were taken? Was a specific pathology selected to put the ROI or were they randomly placed?
Regarding Figure 4, Lines 382-383: At the magnification provided, it is very hard to interpret or visualize the co-localization of the autoradiography signal with lewy bodies and oligodendric inclusions. This may make the results seem unconvincing. It would be helpful if authors provided a high magnification (about 40x) multipanel image. For example, the image could be, Panel A: antibody staining of a ROI, Panel B: ARG of the same ROI at same magnification and scale, Panel C: Co-localized or co-registered image showing both immuno-staining and ARG in the same image. Some of this type of data is shown in figure B4-Line 651, but it is still difficult to visualize co-localization. Also, scale bars would be helpful to identify pathology.
In figure 4, it is not possible to interpret the figure insets and the associated data. Also, scale bars are not visible in those figures.
Methods could be clearly defined for reproducibility:
Line 101: Recombinant fibrils were generated under “constant shaking conditions”. Please specify rpm, shaker make/model used to generate fibrils? Is there any characterization done on the PFFs produced?
Line 102: “ultra centrifugation, resuspended in buffer, sonicated,” details: Please specify centrifugation speeds and sonication details. All these may be important to understanding binding and reproducibility.
Line 253: Please describe immunostaining method in detail. The concentrations of antibody used alongwith their catalog numbers, dilutions etc. to ensure reproduction of results.
Writing suggestion: I looked at the “Cells” instruction for authors and was not sure about the bulleted results version in the current submission. If that is not the preferred writing format for this journal, please submit the results in the usual paragraph style.
Thank you for the opportunity to review.
Author Response
|
Response to Reviewer 1 Comments
|
||
|
1. Summary |
|
|
|
Thank you very much for taking the time to review this manuscript. Please find the detailed responses below and the corresponding revisions/corrections highlighted in red in the re-submitted files.
|
||
|
2. Questions for General Evaluation |
Reviewer’s Evaluation |
Response and Revisions |
|
Does the introduction provide sufficient background and include all relevant references? |
Yes |
We thank the Reviewer for the comments and suggestions that we have tried to address in the revised version of the manuscript. We provide a point-by-point response below. |
|
Are all the cited references relevant to the research? |
Yes |
|
|
Is the research design appropriate? |
Yes |
|
|
Are the methods adequately described? |
Can be improved |
|
|
Are the results clearly presented? |
Must be improved |
|
|
Are the conclusions supported by the results? |
Can be improved |
|
|
3. Point-by-point response to Comments and Suggestions for Authors |
||
|
Comments 1: Most data is presented by quantification of autoradiography data, which in itself can be challenging. In addition, there is a real potential of operator bias in deciding where to put the ROIs. Therefore, more clarity is needed to improve the manuscript. At this stage, the manuscript needs major revision before publication
|
||
|
Response 1: Thank you for pointing this out. We agree with this comment. Therefore, the quantitative autoradiography method description has been revised and ROI delineation figures were added to the Appendix B (Figures B3, B4 and B10).
|
||
|
Comments 2: Figure 5 is missing from the reviewer version, so that data has not been reviewed in this report. Please ensure that it is present in revised version |
||
|
|
||
|
Response 2: Thanks for noticing this. Figure 5 is present in the revised version but was also included in the original version. |
||
|
|
||
|
Comments 3: Another major issue for review are the low resolution figures in the reviewer version. For example, the Scatchard plots in Figure 2 are blurred and un-intelligible. Is there a second binding site in the Scatchard? |
||
|
|
||
|
Response 3: We thank the Reviewer for this comment. The resolution of Figure 2 has been improved. The Scatchard plots suggest only one binding site. |
||
|
|
||
|
Comments 4: It would be helpful to use the global fit analysis to analyze total and non-specific binding at the same time. This helps to visualize the issues with non-specific binding at the same time as total binding. This may be important as a very high concentration of 10uM is used for competition. |
||
|
|
||
|
Response 4: We thank the Reviewer for this comment. The data of the saturation curves have been displayed for both total and non-specific binding in Figure B2. |
||
|
|
||
|
Comments 5: As mentioned before, the authors present substantial quantitative autoradiography analysis. For readers, it would be helpful to detail how the quantification of autoradiography was performed? Please add in supplementary data figures showing where the ROI boxes are placed on representative tissue sections used for quantification. (Line 222). |
||
|
|
||
|
Response 5: We thank the Reviewer for this comment. As we indicated in the manuscript, we delineated the ROIs using toluidine blue staining of adjacent sections to obtain anatomical references for white and grey matter. As mentioned in the reply to comment 1, we have now included additional figures in Appendix B showing the delineation of the ROIs. |
||
|
|
||
|
Comments 6: Similarly, it was unclear how quantification of autoradiography was performed for Appendix B (Line 630). |
||
|
|
||
|
Response 6: For the upper plot shown in Figure B5, the average non-specific binding (fmol/mg) average from all the replicate cores of each donor are expressed as percentage of the respective average total binding. The ROI selection is illustrated in Figure B4. |
||
|
|
||
|
Comments 7: Similarly, in Table 3, specific binding is quantified from several cases and mean±SEM is presented. Please elaborate on how ROI delineation was done, how total and non-specific ROIs were generated in the TMA experiments? Was the entire core a single ROI? In supplementary figures, show representative sections with ROIs to show how measurements were taken? Was a specific pathology selected to put the ROI or were they randomly placed? |
||
|
|
||
|
Response 7: We noticed that Table 3 was uncorrectly numbered in the original submission. It is now numbered correctly as Table 4. The total and non-specific ROI delineation was done for the whole core, except for MSA, where the ROI followed the areas where the binding of [3H]GMC-058 corresponded to α-syn inclusions. For the cores from the substantia nigra of PD patient cases, we selected cores for analysis which contained α-syn inclusion bodies but no A-β deposits of pTau tangles (information which was collected from IHC analysis of adjacent TMA slides). This description is now provided in the revised manuscript (lines 250-258) and the ROIs are presented in Figure B4. |
||
|
|
||
|
Comments 8: Regarding Figure 4, Lines 382-383: At the magnification provided, it is very hard to interpret or visualize the co-localization of the autoradiography signal with lewy bodies and oligodendric inclusions. This may make the results seem unconvincing. It would be helpful if authors provided a high magnification (about 40x) multipanel image. For example, the image could be, Panel A: antibody staining of a ROI, Panel B: ARG of the same ROI at same magnification and scale, Panel C: Co-localized or co-registered image showing both immuno-staining and ARG in the same image. Some of this type of data is shown in figure B4-Line 651, but it is still difficult to visualize co-localization. Also, scale bars would be helpful to identify pathology. |
||
|
|
||
|
Response 8: We thank the Reviewer for this comment and suggestion. We have revised Figure 4 and B7, B8 and B8 in order to increase the resolution and provide a higher magnification to display the correspondence between IHC and ARG. |
||
|
|
||
|
Comments 9: In figure 4, it is not possible to interpret the figure insets and the associated data. Also, scale bars are not visible in those figures. |
||
|
|
||
|
Response 9: In the revised manuscript we have provided Figure 4 with increased resolution and also a scale bar. |
||
|
|
||
|
Comments 10: Methods could be clearly defined for reproducibility: Line 101: Recombinant fibrils were generated under “constant shaking conditions”. Please specify rpm, shaker make/model used to generate fibrils? Is there any characterization done on the PFFs produced? |
||
|
|
||
|
Response 10: These details requested by the reviewer have been included in lines 102-111. |
||
|
|
||
|
Comments 11: Line 102: “ultra centrifugation, resuspended in buffer, sonicated,” details: Please specify centrifugation speeds and sonication details. All these may be important to understanding binding and reproducibility. |
||
|
|
||
|
Response 11: These details requested by the reviewer have been included in lines 107-114. |
||
|
|
||
|
Comments 12: Line 253: Please describe immunostaining method in detail. The concentrations of antibody used alongwith their catalog numbers, dilutions etc. to ensure reproduction of results. |
||
|
|
||
|
Response 12: These details requested by the reviewer have been included in lines 282-295. |
||
|
|
||
|
Comments 13: Writing suggestion: I looked at the “Cells” instruction for authors and was not sure about the bulleted results version in the current submission. If that is not the preferred writing format for this journal, please submit the results in the usual paragraph style. |
||
|
|
||
|
Response 13: We have removed the bullet format of the text. |
||
|
|
||
Reviewer 2 Report
Comments and Suggestions for Authors
The authors developed and characterized a novel radioligand, [³H]GMC-058, intended for in vitro characterization of α-syn pathology. This is an interesting and promising study. However, several concerns remain that should be addressed:
1. The stated aim of the study is to discover and characterize an in vitro radioligand for imaging α-syn pathology. However, [³H]GMC-058 exhibits relatively low binding affinity for α-syn, while demonstrating higher affinity toward tau pathologies, such as those observed in PSP and CBD. Could the authors clarify this point further?
2. The number of brain tissue samples used in some experimental groups appears limited. Do the authors have plans to include additional samples to enhance the statistical robustness of the findings?
3. It would be helpful to know whether the authors have tested [³H]GMC-058 in non-human primate or rodent models of α-synuclein pathology. If so, incorporating such data would further support the translational value of the study.
Author Response
|
Response to Reviewer 2 Comments
|
||
|
1. Summary |
|
|
|
Thank you very much for taking the time to review this manuscript. Please find the detailed responses below and the corresponding revisions/corrections highlighted in red in the re-submitted files.
|
||
|
2. Questions for General Evaluation |
Reviewer’s Evaluation |
Response and Revisions |
|
Does the introduction provide sufficient background and include all relevant references? |
Yes |
We thank the Reviewer for the comments and suggestions that we have tried to address in the revised version of the manuscript. We provide a point-by-point response below. |
|
Are all the cited references relevant to the research? |
Yes |
|
|
Is the research design appropriate? |
Can be improved |
|
|
Are the methods adequately described? |
Yes |
|
|
Are the results clearly presented? |
Can be improved |
|
|
Are the conclusions supported by the results? |
Can be improved |
|
|
3. Point-by-point response to Comments and Suggestions for Authors |
||
|
Comments 1: The stated aim of the study is to discover and characterize an in vitro radioligand for imaging α-syn pathology. However, [³H]GMC-058 exhibits relatively low binding affinity for α-syn, while demonstrating higher affinity toward tau pathologies, such as those observed in PSP and CBD. Could the authors clarify this point further?
|
||
|
Response 1: We thank the Reviewer for this comment. The assays used in vitro to screen the compounds fulfilling selection criteria were performed using the following reference radioligands: [3H]PiB (A-β), [3H]NF-355 (tau), and [3H]-α-syn-tool (α-syn). With regards to α-syn, the assay was performed on recombinant fibrils and it is possible that the affinity of GMC-058 measured using synthetic α-syn fibrils is different from the affinity of the tracer for native α-syn fibrils present in tissue slices or brain homogenates. With regards to tau, it is possible that GMC-058 recognize a different binding site from the one recognised by [3H]NF-355, which has been used for the development of MK-6240, a tau PET radioligand known to bind to 3R/4R tau in AD. This has been discussed in the manuscript (lines 425-431): However, when [3H]GMC-058 was further evaluated in human brain tissue homogenates and sections, it displayed low specific binding in synucleinopathy cases, but clear specific binding in PSP and CBD cases. This observation was unexpected considering that the Ki of GMC-058 measured using recombinant α-syn fibrils was 22.5 nM and in tau-enriched AD tissue was > 200 nM. It is possible that the affinity of GMC-058 measured using recombinant α-syn fibrils was different from the affinity of [3H]GMC-058 to native α-syn fibrils in tissue slices or brain homogenates, resulting in a discrepancy between the two different assay results. With regards to tau, since the competition studies were performed using the reference tau tracer [3H]NF-355, it is possible that GMC-058 recognizes a different binding site on tau fibrils.
|
||
|
Comments 2: The number of brain tissue samples used in some experimental groups appears limited. Do the authors have plans to include additional samples to enhance the statistical robustness of the findings? |
||
|
|
||
|
Response 2: We thank the Reviewer for this comment. Unfortunately, we do not have additional samples that can be included in the analysis. |
||
|
|
||
|
Comments 3: It would be helpful to know whether the authors have tested [³H]GMC-058 in non-human primate or rodent models of α-synuclein pathology. If so, incorporating such data would further support the translational value of the study |
||
|
|
||
|
Response 3: We thank the Reviewer for this comment. Unfortunately, we do not have available tissue from non-human primate or rodent models of α-synuclein pathology. |
||
Round 2
Reviewer 2 Report
Comments and Suggestions for Authors
The authors have addressed all my concerns. I have no further comments.